

# Universal aspects of high-temperature relaxation dynamics in random spin models

Tian-Gang Zhou[1], Wei Zheng[2,3,4] and Pengfei Zhang[4,5,6*]

**1** Institute for Advanced Study, Tsinghua University, Beijing, 100084, China
**2** Hefei National Research Center for Physical Sciences at the Microscale and
School of Physical Sciences, University of Science and Technology of China,
Hefei 230026, China
**3** CAS Center for Excellence in Quantum Information and Quantum Physics,
University of Science and Technology of China, Hefei 230026, China
**4** Hefei National Laboratory, Hefei 230088, China
**5** Department of Physics & State Key Laboratory of Surface Physics,
Fudan University, Shanghai, 200438, China
**6** Shanghai Qi Zhi Institute, AI Tower, Xuhui District, Shanghai 200232, China

⋆ pengfeizhang.physics@gmail.com

## Abstract

Universality is a crucial concept in modern physics, allowing us to capture the essential features of a system's behavior using a small set of parameters. In this work, we unveil universal spin relaxation dynamics in anisotropic random Heisenberg models with infinite-range interactions at high temperatures. Starting from a polarized state, the total magnetization can relax monotonically or decay with long-lived oscillations, determined by the sign of a universal single function $A = -\xi_1^2 + \xi_2^2 - 4\xi_2\xi_3 + \xi_3^2$. Here $(\xi_1, \xi_3, \xi_3)$ characterizes the anisotropy of the Heisenberg interaction. Furthermore, the oscillation shows up only for $A > 0$, with frequency $\Omega \propto \sqrt{A}$. This result is derived from the Kadanoff-Baym equation under the melon diagram approximation, which is consistent with numerical solutions. Furthermore, we verify our theory and approximation using exact diagonalization, albeit for a small system size of $N = 8$. Our study sheds light on the universal aspect of quantum many-body dynamics beyond low energy limit.

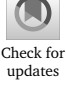

# 1 Introduction

A complete description of realistic many-body systems always contains a large number of parameters. For example, typical solid-state material contains complicated interactions between electrons, phonons, nuclei, and impurities. However, properties that are of physical interest can usually be captured by simple toy models with few parameters. This is a remarkable consequence of universality. The universality states that microscopically different systems can share the same physics at large scales. It usually emerges in long wave length or low-energy limit. For example, phase transitions of many-body systems can be classified into universality classes determined only by the symmetry and dimension of systems [1, 2]. Low-energy scattering between atoms can be well described by a single parameter, the scattering length $a_s$, despite details of underlying microscopic interaction potentials [3]. Aiming at deepening our understanding of realistic systems, discovering new universalities becomes an important subject in modern many-body physics.

Recent years have witnessed a great breakthrough in understanding real time dynamics or relaxation in quantum many-body systems both theoretically [4–30] and experimentally [31–37]. In the previous studies on relaxation, most universal dynamical behaviors emerge in the low temperature or long time scale. That reflects the microscopic details of models are smoothed out in the low energy scale. However at high temperature and short time, it is common believed that most microscopic details are involved in the evolution. Such that the evolution is highly model dependent and hard to observe a universal dynamics. In this work, we unveil that a universal aspect of relaxation dynamics which shows up in an anisotropic Heisenberg model with all-to-all interactions even at high temperatures and short time. The Hamiltonian reads:

$$\hat{H} = \sum_{1 \leq i < j \leq N} J_{ij}(\xi_1 \hat{S}_i^x \hat{S}_j^x + \xi_2 \hat{S}_i^y \hat{S}_j^y + \xi_3 \hat{S}_i^z \hat{S}_j^z) - h(t) \sum_{1 \leq i \leq N} \hat{S}_i^x . \tag{1}$$

This model with different anisotropy parameters $(\xi_1, \xi_2, \xi_3)$ has been realized in cold molecules [38, 39], NV centers [40, 41], trapped fermions [42], Rydberg atoms [43, 44], high spin atoms [45], and solid-state NMR systems [46, 47]. A schematic figure is presented in Fig. 1 (a). Because of random locations or complicated spatial wavefunctions of spin carriers, $J_{ij}$ is usually modeled as independent random Gaussian variables with expectation $\overline{J_{ij}} = \bar{J}/N$ and variance $\overline{\delta J_{ij}^2} = 4J^2/N$.

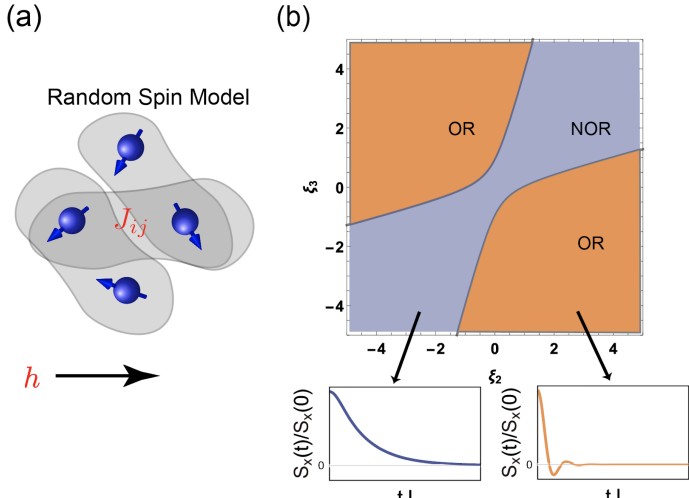

Figure 1: (a). Schematics of the random spin model with random (anisotropic) Heisenberg interactions $J_{ij}$ in the magnetic field $h$. (b). Different dynamical behaviors of the system for different anisotropy parameters $(1, \xi_2, \xi_3)$. The boundary line is determined by $A = -\xi_1^2 + \xi_2^2 - 4\xi_2\xi_3 + \xi_3^2 = 0$, which is symmetric under the reflection along $\xi_2 = \pm\xi_3$. OR and NOR denote the oscillating regime and non-oscillating relaxation regime respectively, distinguished by features of the magnetization relaxation process.

We focus on the following protocol: The system is prepared at high temperatures with a polarization field $h(t < 0) = h$, which induces a magnetization in the $x$ direction. We then monitor the relaxation of the total magnetization after turning off $h$ suddenly at $t = 0$. We find the total magnetization decays either monotonically or with long-lived oscillations, depending on $A = -\xi_1^2 + \xi_2^2 - 4\xi_2\xi_3 + \xi_3^2$. The oscillation only appears for $A > 0$, in which case the frequency satisfies $\Omega \propto J\sqrt{A}$. Importantly, this phenomenon should be understood as a universal property of the relaxation dynamics since the criterion only contains a specific combination of anisotropic parameters, instead of full details of the microscopic model (1). The result is derived from the Kadanoff-Baym equation under the melon diagram approximation, which is consistent with numerical solutions. We further verify our theory and approximation using exact diagonalization, albeit for a small system size of $N = 8$. Our work also provides a novel theoretical framework to analyze the dynamics of randomly interacting quantum spin models.

## 2 Kadanoff-Baym equation

We are interested in the relaxation dynamics of total magnetization. Our theoretical analysis is based on the path-integral approach on the Keldysh contour, as elaborated in [48, 49]. To begin with, we observe that the random spin model can be written in terms of Abrikosov fermion operators $\hat{c}_{i,s}$ with spins $s = \uparrow, \downarrow$ in the single occupation subspace. Explicitly, we have $\hat{S}_i^\alpha = \sum_{ss'} \frac{1}{2}\hat{c}_{i,s}^\dagger (\sigma^\alpha)_{ss'}\hat{c}_{i,s'}$, where $\alpha = x, y, z$ and $\sigma^\alpha$ denote the corresponding Pauli matrices. Since the Hamiltonian (1) exhibits $\pi$ rotation symmetries along the $x$ axis, the total magnetization can only be along the $x$ axis. We thus introduce $m(t) \equiv \langle \hat{S}^x(t) \rangle$. Since the total magnetization is always along the $x$ direction, the magnetization can be computed by real-time Green's functions of fermion operators:

$$m(t) = -iG_{\uparrow\downarrow}^>(t, t) = -iG_{\uparrow\downarrow}^<(t, t), \tag{2}$$

where we have defined

$$
G_{ss'}^>(t_1, t_2) \equiv -i \sum_l \left\langle c_{l,s}(t_1) c_{l,s'}^\dagger(t_2) \right\rangle / N,
$$
$$
G_{ss'}^<(t_1, t_2) \equiv i \sum_l \left\langle c_{l,s'}^\dagger(t_2) c_{l,s}(t_1) \right\rangle / N.
$$
(3)

The relaxation dynamics of $m(t)$ can then be computed once we obtain the Green's functions $G^\gtrless(t_1, t_2)$. It is known that the evolution of $G^\gtrless(t_1, t_2)$ is governed by the Kadanoff-Baym equation, which can be derived by the Schwinger-Dyson equation on the Schwinger-Keldysh contour.

$$
i\partial_{t_1} G^\gtrless + \frac{1}{2} h_{\text{eff}}(t_1) \sigma^x G^\gtrless = \Sigma^R \circ G^\gtrless + \Sigma^\gtrless \circ G^A,
$$
$$
-i\partial_{t_2} G^\gtrless + \frac{1}{2} h_{\text{eff}}(t_2) G^\gtrless \sigma^x = G^R \circ \Sigma^\gtrless + G^\gtrless \circ \Sigma^A.
$$
(4)

Here we have introduced self-energies $\Sigma^\gtrless$ and $\Sigma^{R/A}$. We define the operation $\circ$ for functions with two time variables as $f \circ g \equiv \int dt_3 \, f(t_1, t_3) g(t_3, t_2)$. The retarded and advanced Green's functions $G^{R/A}$ are related to $G^\gtrless$ by $G^{R/A} = \pm\Theta(\pm t_{12})(G^> - G^<)$, where $\Theta(t)$ is the Heaviside step function. Similar relations work for self-energies $\Sigma^{R/A}$. $h_{\text{eff}}(t) = h(t) + \bar{J} m(t)$ is the effective magnetic field, which includes the mean-field contribution from $\bar{J}$. For $t < 0$, the system is prepared in thermal equilibrium. Consequently, we have $G^\gtrless(t_1, t_2) = G_\beta^\gtrless(t_{12})$ for $t_1, t_2 < 0$. For either $t_1 > 0$ or $t_2 > 0$, the Green's functions evolve due to the quantum quench and should be obtained by solving Eq. (4) after the self-energy is specified.

The approximation comes in when we try to relate the self-energies to Green's functions. After transforming into the Abrikosov fermion representation, the random Heisenberg interaction takes the form of random fermion scatterings. Interestingly, such random interaction terms is a close analog of the celebrated complex Sachdev-Ye-Kitaev (SYK) model [50–56]. Motivated by this observation, here we make the melon diagram approximation for the fermion self-energy. A formal argument to control errors is to generalize the Hamiltonian (1) into large-$M$ spins, as in the seminal work by Sachdev and Ye [57]. We promote the original model by adding an additional $M$ indices as

$$
\hat{H} = \frac{1}{\sqrt{M}} \sum_{i<j,\alpha\gamma} J_{ij} \xi^\alpha \hat{T}_i^{\alpha,\gamma} \hat{T}_j^{\alpha,\gamma} - h \sum_i \hat{S}_i^x,
$$
(5)

where we have introduced

$$
\hat{T}_i^{\alpha,\gamma} = \frac{1}{2} \sum_{s_i, m_i} \hat{c}_{i,s_1,m_1}^\dagger (\sigma^\alpha)_{s_1 s_2} (T^\gamma)_{m_1 m_2} \hat{c}_{i,s_2,m_2}, \quad \text{with } \gamma \in \{1, 2, \ldots, M^2 - 1\},
$$

labeling the generators of the $SU(M)$ group. It is known that they satisfy the completeness relation $\sum_\gamma T_{m_1 m_2}^\gamma T_{m_3 m_4}^\gamma = \delta_{m_1 m_4} \delta_{m_2 m_3} - \frac{1}{M} \delta_{m_1 m_2} \delta_{m_3 m_4}$. The external field $h$ only couples to the $SU(2)$ part. The constrain is also promoted as $\sum_{s,m} \hat{c}_{i,s,m}^\dagger \hat{c}_{i,s,m} = M$. Firstly, we take the imaginary time approach in the large-$N$ and large-$M$ limits. The constrain is satisfied automatically due to the particle-hole symmetry, and the self-energy can be obtained by the melon diagrams as in [57]. Finally, this leads to

$$
\Sigma^\gtrless(t_1, t_2) = \frac{J^2}{4} \sum_{\alpha, \alpha'} \xi_\alpha \xi_{\alpha'} \sigma^{\alpha'} G^\gtrless(t_1, t_2) \sigma^\alpha \text{Tr}\left[ \sigma^{\alpha'} G^\gtrless(t_1, t_2) \sigma^\alpha G^\lessgtr(t_2, t_1) \right].
$$
(6)

Here, we omit spin indices for conciseness. We have introduced the anisotropy vector $\xi = (\xi_1, \xi_2, \xi_3)$. The melon diagram approximation may fail in the low-temperature limit

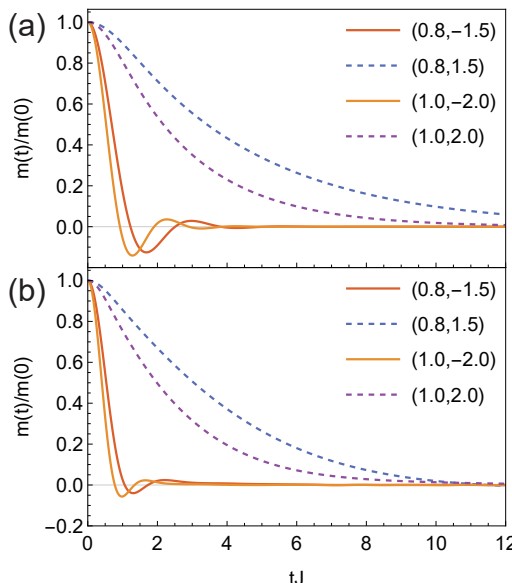

Figure 2: The numerical result for the evolution of the magnetization $m(t)$ by numerically solving: (a) the Kadanoff-Baym equation (4) and (b) exact diagonalization with system size $N = 8$. Initially, the system is in thermal equilibrium with $\beta J = 0.04$, $\bar{J} = 0$ and $h/J = 10$. We take $\xi_1 = 1$ and consider four different anisotropy parameters $(\xi_2, \xi_3) = (0.8, -1.5), (0.8, 1.5), (1, -2)$, and $(1, 2)$, which corresponds to $A = 6.69, -2.91, 12$, and $-4$. The results show that the relaxation of $m(t)$ is monotonic/oscillating if $A < 0/A > 0$. These two numerical results match each other to good precision despite a small $N$.

if the system exhibits spin glass orders [58]. In this work, we avoid this problem by focusing on the high-temperature regime with $\beta J \ll 1$. Combining Eq. (2), (4), and (6) leads to a set of closed equations which determines the relaxation of the magnetization. For later convenience, we also provide matrix elements of Eq. (6) explicitly after using the symmetry of Green's function in Appendix C. Leaving the details in the Appendix. D, we have

$$\Sigma^{\gtrless} = \frac{J^2}{2} \begin{pmatrix} -\xi^2 G_{\uparrow\uparrow}^{\gtrless}(t_1,t_2)^3 + A G_{\uparrow\uparrow}^{\gtrless}(t_1,t_2) G_{\uparrow\downarrow}^{\gtrless}(t_1,t_2)^2 & A G_{\uparrow\uparrow}^{\gtrless}(t_1,t_2)^2 G_{\uparrow\downarrow}^{\gtrless}(t_1,t_2) + \xi^2 G_{\uparrow\downarrow}^{\gtrless}(t_1,t_2)^3 \\ A G_{\uparrow\uparrow}^{\gtrless}(t_1,t_2)^2 G_{\uparrow\downarrow}^{\gtrless}(t_1,t_2) + \xi^2 G_{\uparrow\downarrow}^{\gtrless}(t_1,t_2)^3 & -\xi^2 G_{\uparrow\uparrow}^{\gtrless}(t_1,t_2)^3 + A G_{\uparrow\uparrow}^{\gtrless}(t_1,t_2) G_{\uparrow\downarrow}^{\gtrless}(t_1,t_2)^2 \end{pmatrix}, \quad (7)$$

with $A = -\xi_1^2 + \xi_2^2 - 4\xi_2\xi_3 + \xi_3^2$ and $\xi^2 \equiv \xi_1^2 + \xi_2^2 + \xi_3^2$.

Typical numerical results for $m(t)$ obtained by two methods are shown in Fig. 2. Here we consider examples with $\xi_1 = 1$ and $(\xi_2, \xi_3) = (0.8, -1.5), (0.8, 1.5), (1, -2)$, and $(1, 2)$. We set the initial temperature $\beta J = 0.04$, the polarization field $h/J = 10$ and $\bar{J} = 0$. In the long-time limit, the system exhibits the quantum thermalization to the thermal ensemble with $h = 0$. In this case, $\pi$ rotations along $y$ or $z$ also become the symmetry of the Hamiltonian, which makes $m(\infty) = 0$. According to the relaxation process, different anisotropy parameters can be divided into two groups, under which $m(t)$ relaxes monotonically (for $(\xi_2, \xi_3) = (0.8, 1.5)$ and $(1, 2)$) or with long-lived oscillations (for $(\xi_2, \xi_3) = (0.8, -1.5)$ and $(1, -2)$). Furthermore, we numerically checked that the presence of the oscillation is stable against deformations of parameters. As a result, we propose the Hamiltonian (1) with different $(\xi_1, \xi_2, \xi_3)$ can be separated in parameter regimes with oscillating relaxation (OR) versus non-oscillating relaxation (NOR), as shown in Fig. 1. In Appendix A, we verify that the difference in the dynamical behavior can not be detected in equilibrium via spin susceptibility.

# 3 Oscillation versus monotonic decay

After evolving for a long time, the total magnetization, as well as off-diagonal components of Green's functions, becomes very small. Consequently, we can perform a linearized analysis of the KB equation to reveal the mechanism for the oscillation and determine the criterion for different dynamical behaviors. A differential equation that governs the long-time evolution of the magnetization can be derived following a few steps:

**Step 1.–** The linearized analysis can be largely simplified after the Keldysh rotation. We introduce the standard Keldysh Green's function of fermions as $G^K = G^> + G^<$. The total magnetization can be expressed as its off-diagonal component:

$$m(t) = -iG^K_{\uparrow\downarrow}(t,t)/2\,. \tag{8}$$

We can further combine equations in (4) to derive the equation for $G^K$. On the Keldysh contour, the Schwinger-Dyson equation reads

$$\begin{pmatrix} G^R & G^K \\ 0 & G^A \end{pmatrix}^{-1} = \begin{pmatrix} G^R_0 & 0 \\ 0 & G^A_0 \end{pmatrix} - \begin{pmatrix} \Sigma^R & \Sigma^K \\ 0 & \Sigma^A \end{pmatrix}. \tag{9}$$

Here, we have $\Sigma^K = \Sigma^> + \Sigma^<$. Taking the Keldysh component of Eq. 9, we find

$$G^K = G^R \circ \Sigma^K \circ G^A\,. \tag{10}$$

**Step 2.–** We linearize Eq. (10) around the equilibrium solution in the long-time limit after the quantum thermalization. We expand $G^a(t_1, t_2) = G^{a,\beta_f}(t_{12}) + \delta G^a(t_1, t_2)$, where $G^{a,\beta_f}(t)$ is the equilibrium Green's function on the final state. Leaving the details in the appendix. D, the off-diagonal element of (10) reads

$$\begin{aligned} \delta G^K_{\uparrow\downarrow} &= G^{R,\beta_f}_{\uparrow\uparrow} \circ \delta\Sigma^K_{\uparrow\downarrow} \circ G^{A,\beta_f}_{\uparrow\uparrow}\,, \\ \delta\Sigma^K_{\uparrow\downarrow} &= \frac{1}{4}J^2 A\left((G^{>,\beta_f}_{\uparrow\uparrow})^2 + (G^{<,\beta_f}_{\uparrow\uparrow})^2\right)\delta G^K_{\uparrow\downarrow}\,, \end{aligned} \tag{11}$$

where we have used the fact that the equilibrium state contains no magnetization, and is approximately at infinite temperature. The second fact leads to $\Sigma^{K,\beta_f} \approx 0$. Since $G^{K,\beta_f}_{\uparrow\downarrow} = 0$, Eq. (8) is equivalent to $m(t) = -i\delta G^K_{\uparrow\downarrow}(t,t)/2$.

**Step 3.–** To proceed, we need to obtain approximations for $G^{a,\beta_f}_{\uparrow\uparrow}$. In thermal equilibrium with $h = 0$, the self-energies (6) can be simplified as

$$\Sigma^{\gtrless,\beta_f}_{ss'}(t) = -\frac{J^2\xi^2}{2}G^{\gtrless,\beta_f}_{ss}(t)^3\delta_{ss'}\,, \tag{12}$$

where we have used $G^{>,\beta_f}_{ss}(t) = -G^{<,\beta_f}_{ss}(-t)$ due to the particle-hole symmetry. Eq. (12) then matches the self-energy of the Majorana $SYK_4$ model with effective coupling constant $J|\xi|/\sqrt{2}$. It is known that at high temperatures $\beta J \ll 1$, the SYK model can be described by weakly interacting quasi-particles [59]. Taking the Lorentzian approximation, we have

$$G^{R/A,\beta_f}_{\uparrow\uparrow}(t) \approx \mp i\Theta(\pm t)e^{-\Gamma|t|/2}\,, \qquad G^{\gtrless,\beta_f}_{\uparrow\uparrow}(t) \approx \mp ie^{-\Gamma|t|/2}/2\,, \tag{13}$$

with quasi-particle decay rate $\Gamma \propto J$.

**Step 4.–** Using the high-temperature solution, we get

$$\begin{aligned} \delta G^K_{\uparrow\downarrow}(t_1, t_4) &= \int dt_2\, dt_3\, G^R_{\uparrow\uparrow}(t_1, t_2)\left(-\frac{1}{8}J^2 A e^{-|t_2-t_3|\Gamma}\delta G^K_{\uparrow\downarrow}(t_2, t_3)\right)G^A_{\downarrow\downarrow}(t_3, t_4) \\ &= -\frac{1}{8}J^2 A\int dt_2\, dt_3\, e^{-\frac{\Gamma}{2}(t_1-t_2)}\Theta(t_1-t_2)e^{-|t_2-t_3|\Gamma}\delta G^K_{\uparrow\downarrow}(t_2, t_3)e^{\frac{\Gamma}{2}(t_3-t_4)}\Theta(-t_3+t_4)\,. \end{aligned} \tag{14}$$

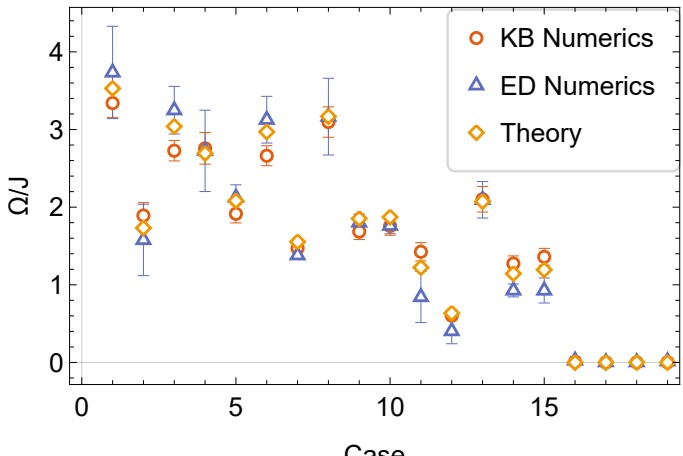

Figure 3: A comparison between the theoretical prediction $\Omega_{Th}/J = c_0\sqrt{A}$ and numerical simulations. Here we choose $c_0 = 1$. In each case, we randomly choose the anisotropic parameter $(\xi_1, \xi_2, \xi_3)$. Initially, the system is also in thermal equilibrium with $\beta J = 0.04$, $\bar{J} = 0$, and $h/J = 10$. The numerical data are obtained by fitting numerics based on the KB equation and ED method. The error bars correspond to standard deviations when concerning the different fitting regions.

Multiply $e^{\frac{\Gamma}{2}t_1}$ and take $\partial_{t_1}$ on the Eq. (14), which gives

$$
\begin{aligned}
\partial_{t_1}\left[e^{\frac{\Gamma}{2}t_1}\delta G^K_{\uparrow\downarrow}(t_1,t_4)\right] &= -\frac{1}{8}J^2 A\int \mathrm{d}t_2\,\mathrm{d}t_3\, e^{\frac{\Gamma}{2}t_2}\delta(t_1-t_2)e^{-|t_2-t_3|\Gamma}\delta G^K_{\uparrow\downarrow}(t_2,t_3)e^{\frac{\Gamma}{2}(t_3-t_4)}\Theta(-t_3+t_4) \\
&= -\frac{1}{8}J^2 A\int \mathrm{d}t_3\, e^{\frac{\Gamma}{2}t_1}e^{-|t_1-t_3|\Gamma}\delta G^K_{\uparrow\downarrow}(t_1,t_3)e^{\frac{\Gamma}{2}(t_3-t_4)}\Theta(-t_3+t_4).
\end{aligned}
\tag{15}
$$

Then multiply $e^{\frac{\Gamma}{2}t_4}$ and again take $\partial_{t_4}$ on both sides of Eq. (15), which leads to a differential equation

$$
\left(\partial_{t_1}+\frac{\Gamma}{2}\right)\left(\partial_{t_2}+\frac{\Gamma}{2}\right)\delta G^K_{\uparrow\downarrow} = -\frac{A}{8}J^2 e^{-\Gamma|t_{12}|}\delta G^K_{\uparrow\downarrow}.
\tag{16}
$$

Eq. (16) is the starting point for analyzing the relaxation dynamics. Since it is invariant under time translations, we separate out the center-of-mass time dependence by introducing $\delta G^K_{\uparrow\downarrow}(t_1,t_2) = \mathrm{Re}\, e^{-\lambda\frac{t_1+t_2}{2}}\varphi(t_{12})$. The relaxation is oscillatory only if $\lambda$ is complex. Interestingly, $\varphi(t_{12})$ then satisfies the 1D Schrödinger equation

$$
-\frac{(\Gamma-\lambda)^2}{4}\varphi(t_{12}) = -\partial^2_{t_{12}}\varphi(t)+\frac{A}{8}J^2 e^{-\Gamma|t_{12}|}\varphi(t_{12}),
\tag{17}
$$

where $-\frac{(\Gamma-\lambda)^2}{4}$ plays the role of the energy $E$ and $\frac{A}{8}J^2 e^{-\Gamma|t_{12}|}$ plays the role of potential $V$. Eq. (17) suggests the boundary line between the oscillating regime and the non-oscillating regime is at $A = 0$: For $A < 0$, the potential energy is negative. It is known that in 1D any attractive potential exhibits at least one bound state. Denoting the energy of the ground state as $-|E_0|$, we can solve $\lambda = \Gamma - 2\sqrt{|E_0|}$, which is real. Consequently, we expect the magnetization relaxes monotonically. For $A > 0$, the potential is repulsive. The eigenstates of the (17) are scattering modes with continuous positive energy $E$. We find $\lambda = \Gamma \pm 2i\sqrt{E}$, which is complex. This leads to oscillations in the relaxation process.

To further determine the typical oscillation frequency $\Omega$, we need to determine the typical energy $E$ that contributes to the quench dynamics. According to Eq. (8), the magnetization probes the decay of the wave function at $t_{12} = 0$, where the potential energy is $\sim AJ^2$.

For $E \ll AJ^2$, the eigenstate has exponentially small weight near $t_{12} = 0$. As a result, the corresponding contribution to $m(t)$ can be neglected. We can approximate

$$m(t) \sim \int_{AJ^2} dE \, c(E) e^{-\Gamma t - 2i\sqrt{E}t} \,. \tag{18}$$

Here $c(E)$ is some smooth function determined by the initial condition. We then expect $\Omega \approx c_0 \sqrt{A}J$, with some $O(1)$ constant $c_0$ which does not depend on parameters in the Hamiltonian (1) and should be extracted using numerics. Interestingly, the result predicts the oscillation period $T = 2\pi/\Omega$ diverges as we approach $A = 0$, which can be viewed as an analog of the divergence of the correlation length in traditional phase transition described by order parameters.

We comment that our results unveil the universality of relaxation dynamics in random spin models. Although the microscopic model in Eq. (1) contains several parameters, the criterion for the different relaxation behaviors, as well as the oscillation frequency, only depends on a specific combination $A$. This is a direct analog of universality in the scattering theory, where for a complicated potential, the low-energy scattering problem can only depend on a specific combination of microscopic parameters, which is the scattering length.

We further compare our prediction of the oscillation frequency $\Omega \approx c_0 \sqrt{A}J$ to numerical results. We obtain $\Omega$ in numerics by fitting $m(t) = m_0 \cos(\Omega t + \theta)e^{-\Gamma t} + m_{\text{offset}}$. Here $m_0$ is the amplitude, $\theta$ is the phase, $\Gamma$ is the quasi-particle decay rate, and $m_{\text{offset}}$ is the offset which is significant in the finite $N$ ED numerics. The fitting particularly focuses on the matching in the small $m(t)$ region. Hence the detailed fitting region and the error bars caused by such ambiguity are left to the Appendix. E. The results are shown in Fig. 3. We randomly choose the anisotropic parameters, and the first 15 cases correspond to $A > 0$, and the last 4 cases to $A < 0$ (see Appendix. E). Among the $A > 0$ cases, the mean ratio between the numerical data and the polynomial $A$ reads $\overline{\Omega_{\text{KB}}/(J\sqrt{A})} = 0.995 \pm 0.018$ and $\overline{\Omega_{\text{ED}}/(J\sqrt{A})} = 0.94 \pm 0.04$. Therefore, we set $c_0 = 1$ for theoretical predictions in Fig. 3. Although the error bars for ED numerics are significantly larger than KB numerics since the calculation is based on the finite $N = 8$ system, we find the theoretical prediction of the oscillation frequency almost matches the KB results and the ED results, up to the error bars. From Fig. 3, most notably, the OR and NOR relaxation are sharply distinguished by the $A > 0$ or $A < 0$ criterion, which is perfectly aligned with our theoretical analysis.

## 4 Discussions

In this work, we show that the random Heisenberg model with all-to-all interactions exhibits universal relaxation dynamics governed by a single parameter $A = -\xi_1^2 + \xi_2^2 - 4\xi_2\xi_3 + \xi_3^2$. Unlike traditional examples where the universality emerges in the low-energy limit, here the universal physics appears at high temperatures. For $A < 0$, the magnetization decays monotonically after we turn off the polarization field. For $A > 0$, long-lived oscillation appears during the relaxation process, with a frequency $\Omega \propto J\sqrt{A}$. Our theoretical analysis is based on the path-integral approach along the Keldysh contour, using the KB equation. This includes both perturbative analysis and numerical solutions, further validated by simulations using ED.

We remark that quantum coherence is essential for the existence of the oscillating relaxation regime. As an example, if we spoil the coherence by considering time-dependent random interactions instead of static interactions, the magnetization is expected to decay monotonically: After replacing $J_{ij}$ with Brownian variables $J_{ij}(t)$, Eq. (16) is replaced by

$$(\partial_t + \Gamma)\delta G_{\uparrow\downarrow}^K(t,t) = -\frac{AJ}{8}\delta G_{\uparrow\downarrow}^K(t,t), \tag{19}$$

as derived in the Appendix B. This results in $m(t) \sim e^{-(\Gamma+AJ/8)t}$ with a simple exponential decay, on contrary to the existence of different dynamical behaviors in the static case.

We also point out that amazingly our criteria $A > 0$ for the oscillation regime matches the criteria proposed in [56] for the presence of the instability towards the formation of wormholes with $\xi_1 = \xi_2 = 1$. However, the analysis in [56] focuses on the low-temperature regime, while in this work we focus on high temperatures. This makes it difficult to establish a direct relationship between the two theoretical analyses. It would be interesting to explore whether there is some form of duality between the high-temperature and low-temperature limits. Given that the wormhole phase is non-chaotic, it would also be intriguing to study the out-of-time-order correlator or the operator size distribution in regimes with different dynamical behaviors. Experimentally, our results can be readily verified through quantum quench experiments in NMR systems.

*Note Added.* Universal behaviors of auto-correlation function related to the quench dynamics discussed here, including oscillatory versus non-oscillatory behavior, have been related to Lanczos coefficients computed for determining the Krylov complexity in Ref. [60].

## Acknowledgments

We are especially grateful to the invaluable discussions with Hui Zhai, whose advice is indispensable for the whole work. We thank Riqiang Fu, Yuchen Li, Xinhua Peng, Xiao-Liang Qi and Ren Zhang for their helpful discussions.

**Funding information** This project is supported by the Shanghai Rising-Star Program under grant number 24QA2700300, the NSFC under grant 12374477, and the Innovation Program for Quantum Science and Technology 2024ZD0300101.

## A  Susceptibility in thermal equilibrium

As we discussed in the main text, we calculate the equilibrium susceptibility. We take a small external magnetic field in the $x$ direction in the equilibrium thermal state. The finite difference susceptibility is defined as $\chi = \langle \hat{S}_x \rangle_h / h$, where $\langle \hat{S}_x \rangle_h$ means the thermal average in the external magnetic field $h$. First, we find the exact diagonalization and large-$N$ Kadanoff-Baym results in Fig. 4 agree well with each other. Second, equilibrium susceptibility in Fig. 4 is highly in contrast with the criterion for the relaxation dynamics $A$ in Fig. 1 (b). There is no appreciable distinction between $A > 0$ and $A < 0$ region correspondingly in the plot of the equilibrium. Hence, it reveals the significance of our dynamical framework.

## B  Analysis for Brownian interactions

To check the effect of coherence, we place $J_{ij}$ in the Hamiltonian Eq. (5) with Brownian variables $J_{ij}(t)$, where $\overline{J_{ij}(t_1)J_{ij}(t_2)} = 4J^2/N\delta(t_1 - t_2)$. The self energy is replaced by

$$
\begin{aligned}
\Sigma^{\gtrless}(t_1, t_2) = \frac{1}{2}J^2 \delta(t_1 - t_2) \\
\times \begin{pmatrix} -\xi^2 G_{\uparrow\uparrow}^{\gtrless}(t_1,t_2)^3 + AG_{\uparrow\uparrow}^{\gtrless}(t_1,t_2)G_{\uparrow\downarrow}^{\gtrless}(t_1,t_2)^2 & AG_{\uparrow\uparrow}^{\gtrless}(t_1,t_2)^2 G_{\uparrow\downarrow}^{\gtrless}(t_1,t_2) + \xi^2 G_{\uparrow\downarrow}^{\gtrless}(t_1,t_2)^3 \\ AG_{\uparrow\uparrow}^{\gtrless}(t_1,t_2)^2 G_{\uparrow\downarrow}^{\gtrless}(t_1,t_2) + \xi^2 G_{\uparrow\downarrow}^{\gtrless}(t_1,t_2)^3 & -\xi^2 G_{\uparrow\uparrow}^{\gtrless}(t_1,t_2)^3 + AG_{\uparrow\uparrow}^{\gtrless}(t_1,t_2)G_{\uparrow\downarrow}^{\gtrless}(t_1,t_2)^2 \end{pmatrix}.
\end{aligned}
\tag{B.1}
$$

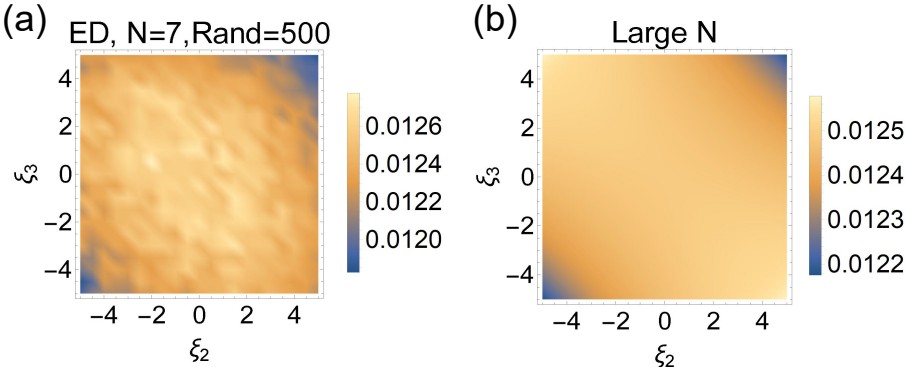

Figure 4: (a) The susceptibility of the exact diagonalization numerics. We choose site number $N = 7$, and random realizations to be 500. (b) The susceptibility of the large-$N$ numerics. In both cases, to obtain the susceptibility, we choose the equilibrium thermal state with temperature $T/J = 10.0$, and external magnetic field with $h/J = 0.05$ in $x$ direction. We take the region of anisotropic parameter $\xi_2, \xi_3$ to be $[-5, 5]$, and the discretization step is $\Delta\xi = 0.5$.

Therefore, following the same steps, the linearization of the Schwinger-Dyson equation $G^K = G^R \circ \Sigma^K \circ G^A$ leads to

$$
\begin{aligned}
\delta G^K_{\uparrow\downarrow}(t_1, t_1) &= \left[ G^{R,\beta_f}_{\uparrow\uparrow} \circ \delta\Sigma^K_{\uparrow\downarrow} \circ G^{A,\beta_f}_{\uparrow\uparrow} \right](t_1, t_1), \\
\delta\Sigma^K_{\uparrow\downarrow}(t_1, t_2) &= \frac{1}{4}J^2 A\delta(t_1 - t_2)\left( G^{>,\beta_f}_{\uparrow\uparrow}(t_1, t_1)^2 + G^{<,\beta_f}_{\uparrow\uparrow}(t_1, t_1)^2 \right)\delta G^K_{\uparrow\downarrow}(t_1, t_1).
\end{aligned}
\tag{B.2}
$$

To form close equation groups, we set the time argument of the first equation to be $(t_1, t_1)$, and then we study the equal time perturbation of the Green's function $\delta G^K_{\uparrow\downarrow}(t, t)$.

The equilibrium Green's function still has the form of Eq. (13) with Lorentz approximation. Using such a solution, we get

$$
\begin{aligned}
\delta G^K_{\uparrow\downarrow}(t_1, t_1) &= \int dt_2\, G^R_{\uparrow\uparrow}(t_1, t_2)\left( -\frac{1}{8}J^2 A e^{-|t_2 - t_2|\Gamma}\delta G^K_{\uparrow\downarrow}(t_2, t_2) \right) G^A_{\downarrow\downarrow}(t_2, t_1) \\
&= -\frac{1}{8}J^2 A \int dt_2 e^{-\frac{\Gamma}{2}(t_1 - t_2)}\Theta(t_1 - t_2)\delta G^K_{\uparrow\downarrow}(t_2, t_2)e^{\frac{\Gamma}{2}(t_2 - t_1)}\Theta(-t_2 + t_1) \\
&= -\frac{1}{8}J^2 A \int dt_2 e^{-\Gamma(t_1 - t_2)}\Theta(t_1 - t_2)\delta G^K_{\uparrow\downarrow}(t_2, t_2).
\end{aligned}
\tag{B.3}
$$

Multiply $e^{\Gamma t_1}$ and take $\partial_{t_1}$ on the Eq. (B.3), which gives

$$
\partial_{t_1}\left[ e^{\Gamma t_1}\delta G^K_{\uparrow\downarrow}(t_1, t_1) \right] = -\frac{1}{8}J^2 A \int dt_2\, e^{\Gamma t_2}\delta(t_1 - t_2)\delta G^K_{\uparrow\downarrow}(t_2, t_2).
\tag{B.4}
$$

Finally leads to

$$
(\partial_t + \Gamma)\delta G^K_{\uparrow\downarrow}(t, t) = -\frac{AJ}{8}\delta G^K_{\uparrow\downarrow}(t, t).
\tag{B.5}
$$

Solving this differential equation leads to $\delta G^K_{\uparrow\downarrow}(t, t) \sim m(t) \sim e^{-(\Gamma + AJ/8)t}$ with a simple exponential decay without any oscillation for arbitrary anisotropy parameters.

# C  Symmetry on Green's function

The greater Green's function in Eq. (3) can be explicitly expressed in $2 \times 2$ matrix form.

$$G^{>}_{s_1 s_2}(t_1, t_2) = -i \langle \hat{c}_{s_1}(t_1) \hat{c}^{\dagger}_{s_2}(t_2) \rangle = \begin{pmatrix} G^{>}_{\uparrow\uparrow}(t_1, t_2) & G^{>}_{\uparrow\downarrow}(t_1, t_2) \\ G^{>}_{\downarrow\uparrow}(t_1, t_2) & G^{>}_{\downarrow\downarrow}(t_1, t_2) \end{pmatrix}_{s_1 s_2}. \tag{C.1}$$

There are two symmetries crucial for the later simplification. The first can be regarded as $\pi$ rotation of axis $x$.

$$\begin{aligned} \hat{c}_{s_1} &\to \sum_{s'} (i\sigma^x)_{s_1 s'} \hat{c}_{s'}, \\ \hat{c}^{\dagger}_{s_1} &\to \sum_{s'} \hat{c}^{\dagger}_{s'} (-i\sigma^x)_{s' s_1}. \end{aligned} \tag{C.2}$$

With symmetry in Eq. (C.2), $\{\hat{S}^x, \hat{S}^y, \hat{S}^z\}$ is mapped to $\{\hat{S}^x, -\hat{S}^y, -\hat{S}^z\}$, and therefore keeps Eq. (1) invariant. As a consequence, the symmetry of Green's function leads to

$$\begin{aligned} G^{>}_{s_1 s_2}(t_1, t_2) &\to \sum_{s' s''} \sigma^x_{s_1 s'} G^{>}_{s' s''}(t_1, t_2) \sigma^x_{s'' s_2} \\ &= \begin{pmatrix} G^{>}_{\downarrow\downarrow}(t_1, t_2) & G^{>}_{\downarrow\uparrow}(t_1, t_2) \\ G^{>}_{\uparrow\downarrow}(t_1, t_2) & G^{>}_{\uparrow\uparrow}(t_1, t_2) \end{pmatrix}_{s_1 s_2}. \end{aligned} \tag{C.3}$$

The second symmetry is combined with particle-hole symmetry and rotation, which reads as

$$\begin{aligned} \hat{c}_{s_1} &\to \sum_{s'} (i\sigma^y)_{s_1 s'} \hat{c}^{\dagger}_{s'}, \\ \hat{c}^{\dagger}_{s_1} &\to \sum_{s'} \hat{c}_{s'} (-i\sigma^y)_{s' s_1}. \end{aligned} \tag{C.4}$$

Similarly, with symmetry in Eq. (C.4), $\{\hat{S}^x, \hat{S}^y, \hat{S}^z\}$ is mapped to $\{\hat{S}^x, -\hat{S}^y, \hat{S}^z\}$, and therefore keeps Eq. (1) invariant. Also, the symmetry of Green's function leads to

$$\begin{aligned} G^{>}_{s_1 s_2}(t_1, t_2) &\to -\sum_{s' s''} \sigma^y_{s_1 s'} G^{<}_{s'' s'}(t_2, t_1) \sigma^y_{s'' s_2} \\ &= \begin{pmatrix} -G^{<}_{\uparrow\uparrow}(t_2, t_1) & G^{<}_{\uparrow\downarrow}(t_2, t_1) \\ G^{<}_{\uparrow\downarrow}(t_2, t_1) & -G^{<}_{\uparrow\uparrow}(t_2, t_1) \end{pmatrix}_{s_1 s_2}, \end{aligned} \tag{C.5}$$

where the second line to the third line uses the symmetry obtained in Eq. (C.3). Finally, we can exchange $>$ and $<$ symbols in Eq. (C.3) and Eq. (C.3) to obtain another two symmetry in terms of Green's function.

# D  Derivation of the self-energy

## D.1  Simplification of Eq. (6)

Starting from Eq. (6), we derive the corresponding analytical formula. We first consider the trace part $\text{Tr}\left[\sigma^{\alpha'} G^{\gtrless}(t_1, t_2) \sigma^\alpha G^{\lessgtr}(t_2, t_1)\right]$. In the basis of $\alpha', \alpha = \{x, y, z\}$, direct calculation of the trace part reads

$$\begin{aligned} &\text{Tr}\left[\sigma^{\alpha'} G^{\gtrless}(t_1, t_2) \sigma^\alpha G^{\lessgtr}(t_2, t_1)\right] \\ &= -\text{Tr}\left[\sigma^y \sigma^{\alpha'} G^{\gtrless}(t_1, t_2) \sigma^\alpha \sigma^y G^{\gtrless}(t_1, t_2)\right] \\ &= \begin{pmatrix} -2G^{\gtrless}_{\uparrow\uparrow}(t_1, t_2)^2 + 2G^{\gtrless}_{\uparrow\downarrow}(t_1, t_2)^2 & 0 & 0 \\ 0 & -2G^{\gtrless}_{\uparrow\uparrow}(t_1, t_2)^2 - 2G^{\gtrless}_{\uparrow\downarrow}(t_1, t_2)^2 & 4iG^{\gtrless}_{\uparrow\uparrow}(t_1, t_2)G^{\gtrless}_{\uparrow\downarrow}(t_1, t_2) \\ 0 & -4iG^{\gtrless}_{\uparrow\uparrow}(t_1, t_2)G^{\gtrless}_{\uparrow\downarrow}(t_1, t_2) & -2G^{\gtrless}_{\uparrow\uparrow}(t_1, t_2)^2 - 2G^{\gtrless}_{\uparrow\downarrow}(t_1, t_2)^2 \end{pmatrix}_{\alpha'\alpha}. \end{aligned} \tag{D.1}$$

The second line applies the symmetry Eq. (C.5) and implicitly use Eq. (C.3) to ensure $\left(G^{\gtrless}\right)^T = G^{\gtrless}$. The off-diagonal matrix elements in the first row and column correspond to the $x$ direction, which is nontrivially disappeared. Since the initial external field is in the $x$ direction, such anisotropic is exactly reflected in the components of self-energy by applying the symmetry constraints.

The self-energy composites different components $\alpha, \alpha'$ which represents the internal spin interaction. Therefore we show each non-zero contribution in Eq. (6).

$\alpha' = x, \ \alpha = x$ :

$$\frac{1}{2}J^2\xi_1^2\begin{pmatrix} G_{\uparrow\uparrow}^{\gtrless}(t_1,t_2) & G_{\uparrow\downarrow}^{\gtrless}(t_1,t_2) \\ G_{\uparrow\downarrow}^{\gtrless}(t_1,t_2) & G_{\uparrow\uparrow}^{\gtrless}(t_1,t_2) \end{pmatrix}\left(-G_{\uparrow\uparrow}^{\gtrless}(t_1,t_2)^2 + G_{\uparrow\downarrow}^{\gtrless}(t_1,t_2)^2\right),$$ (D.2)

$\alpha' = y, \ \alpha = y$ :

$$\frac{1}{2}J^2\xi_2^2\begin{pmatrix} -G_{\uparrow\uparrow}^{\gtrless}(t_1,t_2) & G_{\uparrow\downarrow}^{\gtrless}(t_1,t_2) \\ G_{\uparrow\downarrow}^{\gtrless}(t_1,t_2) & -G_{\uparrow\uparrow}^{\gtrless}(t_1,t_2) \end{pmatrix}\left(G_{\uparrow\uparrow}^{\gtrless}(t_1,t_2)^2 + G_{\uparrow\downarrow}^{\gtrless}(t_1,t_2)^2\right),$$ (D.3)

$\alpha' = z, \ \alpha = z$ :

$$\frac{1}{2}J^2\xi_3^2\begin{pmatrix} -G_{\uparrow\uparrow}^{\gtrless}(t_1,t_2) & G_{\uparrow\downarrow}^{\gtrless}(t_1,t_2) \\ G_{\uparrow\downarrow}^{\gtrless}(t_1,t_2) & -G_{\uparrow\uparrow}^{\gtrless}(t_1,t_2) \end{pmatrix}\left(G_{\uparrow\uparrow}^{\gtrless}(t_1,t_2)^2 + G_{\uparrow\downarrow}^{\gtrless}(t_1,t_2)^2\right),$$ (D.4)

$\alpha' = y, \ \alpha = z$ :

$$J^2\xi_2\xi_3\begin{pmatrix} G_{\uparrow\downarrow}^{\gtrless}(t_1,t_2) & -G_{\uparrow\uparrow}^{\gtrless}(t_1,t_2) \\ -G_{\uparrow\uparrow}^{\gtrless}(t_1,t_2) & G_{\uparrow\downarrow}^{\gtrless}(t_1,t_2) \end{pmatrix}\left(G_{\uparrow\uparrow}^{\gtrless}(t_1,t_2)G_{\uparrow\downarrow}^{\gtrless}(t_1,t_2)\right),$$ (D.5)

$\alpha' = z, \ \alpha = y$ :

$$J^2\xi_2\xi_3\begin{pmatrix} G_{\uparrow\downarrow}^{\gtrless}(t_1,t_2) & -G_{\uparrow\uparrow}^{\gtrless}(t_1,t_2) \\ -G_{\uparrow\uparrow}^{\gtrless}(t_1,t_2) & G_{\uparrow\downarrow}^{\gtrless}(t_1,t_2) \end{pmatrix}\left(G_{\uparrow\uparrow}^{\gtrless}(t_1,t_2)G_{\uparrow\downarrow}^{\gtrless}(t_1,t_2)\right).$$ (D.6)

We finally arrive at the full self-energy by collecting all these terms

$$\Sigma^{\gtrless}(t_1,t_2) = \frac{J^2}{2}\begin{pmatrix} -\xi^2 G_{\uparrow\uparrow}^{\gtrless}(t_1,t_2)^3 + A G_{\uparrow\uparrow}^{\gtrless}(t_1,t_2)G_{\uparrow\downarrow}^{\gtrless}(t_1,t_2)^2 & A G_{\uparrow\uparrow}^{\gtrless}(t_1,t_2)^2 G_{\uparrow\downarrow}^{\gtrless}(t_1,t_2) + \xi^2 G_{\uparrow\downarrow}^{\gtrless}(t_1,t_2)^3 \\ A G_{\uparrow\uparrow}^{\gtrless}(t_1,t_2)^2 G_{\uparrow\downarrow}^{\gtrless}(t_1,t_2) + \xi^2 G_{\uparrow\downarrow}^{\gtrless}(t_1,t_2)^3 & -\xi^2 G_{\uparrow\uparrow}^{\gtrless}(t_1,t_2)^3 + A G_{\uparrow\uparrow}^{\gtrless}(t_1,t_2)G_{\uparrow\downarrow}^{\gtrless}(t_1,t_2)^2 \end{pmatrix},$$ (D.7)

where the polynomials $\xi^2 = \xi_1^2 + \xi_2^2 + \xi_3^2$ and $A = -\xi_1^2 + \xi_2^2 - 4\xi_2\xi_3 + \xi_3^2$.

## D.2 Perturbation

With perturbation on the off-diagonal term on the Eq. (10), in principle we have

$$\delta G_{\uparrow\downarrow}^K = \sum_{s_1,s_2}\underbrace{G_{\uparrow s_1}^{R,\beta_f}\circ\delta\Sigma_{s_1 s_2}^K\circ G_{s_2\downarrow}^{A,\beta_f}}_{\text{Eq. (D.8)(a)}} + \underbrace{\delta G_{\uparrow s_1}^R\circ\Sigma_{s_1 s_2}^{K,\beta_f}\circ G_{s_2\downarrow}^{A,\beta_f}}_{\text{Eq. (D.8)(b)}} + \underbrace{G_{\uparrow s_1}^{R,\beta_f}\circ\Sigma_{s_1 s_2}^{K,\beta_f}\circ\delta G_{s_2\downarrow}^A}_{\text{Eq. (D.8)(c)}}.$$ (D.8)

For Eq. (D.8)(a), all off-diagonal components of $G^{R/A,\beta_f}$ are zero, and consequently we only take $s_1 = \uparrow$ and $s_2 = \downarrow$. For Eq. (D.8)(b), the vanishing off-diagonal component of $G^{A,\beta_f}$, $\Sigma^{K,\beta_f}$ requires $s_2 = \downarrow$ and then $s_1 = \downarrow$. However, in the infinite high-temperature region we have $\Sigma_{\downarrow\downarrow}^{K,\beta_f} = 0$

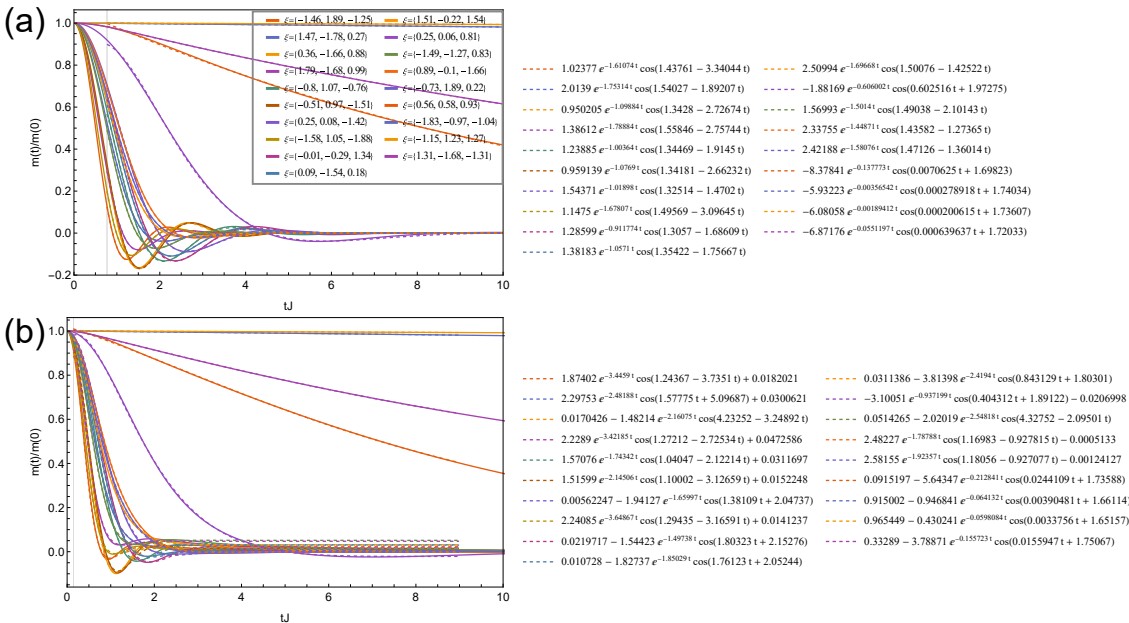

Figure 5: Fitting of (a) the large-$N$ dynamics; (b) the exact diagonalization dynamics corresponding to the main text fig. 3. The solid line is the numerical data, and the dashed line is the fitting curve. The inset legend lists each anisotropic parameter corresponding to the 19 cases in the main text. Besides, the right panel shows the fitting functions for each case.

as a result of the fluctuation-dissipation theorem [48, 49]. In the following content, we can also verify this with the specific ansatz for equilibrium Green's function in Eq. (12), (13). Therefore Eq. (D.8)(b) vanishes, and we can show Eq. (D.8)(c) vanishes with similar arguments. Combining all the argument, only the off-diagonal term in Eq. (D.8)(a) survives and finally leads to Eq. (11).

To obtain $\delta\Sigma^K_{s_1 s_2}$ with $s_1 = \uparrow$ and $s_2 = \downarrow$, we can use the relation $\Sigma^K = \Sigma^> + \Sigma^<$ and the off-diagonal term in Eq. (D.7). Near the equilibrium solution,

$$G^{\gtrless}_{\uparrow\uparrow}(t_1, t_2)^2 G^{\gtrless}_{\uparrow\downarrow}(t_1, t_2) \approx G^{\gtrless, \beta_f}_{\uparrow\uparrow}(t_1, t_2)^2 \delta G^{\gtrless}_{\uparrow\downarrow}(t_1, t_2),$$

and $G^{\gtrless}_{\uparrow\downarrow}(t_1, t_2)^3 \approx \delta G^{\gtrless}_{\uparrow\downarrow}(t_1, t_2)^3$. We will drop the third order contribution in $\delta G^{\gtrless}_{\uparrow\downarrow}$ and only keep the linear order. Finally, it leads to the result

$$
\begin{aligned}
\delta\Sigma^K_{\uparrow\downarrow} &= \frac{1}{2}J^2 A G^{>,\beta_f}_{\uparrow\uparrow}(t_1, t_2)^2 \delta G^>_{\uparrow\downarrow}(t_1, t_2) + \frac{1}{2}J^2 A G^{<,\beta_f}_{\uparrow\uparrow}(t_1, t_2)^2 \delta G^<_{\uparrow\downarrow}(t_1, t_2) \\
&= \frac{1}{4}J^2 A \left( G^{>,\beta_f}_{\uparrow\uparrow}(t_1, t_2)^2 + G^{<,\beta_f}_{\uparrow\uparrow}(t_1, t_2)^2 \right) \delta G^K_{\uparrow\downarrow}(t_1, t_2).
\end{aligned}
$$
(D.9)

# E  Estimation of frequency and error bar in Fig. 3

Fig. 3 in the main text shows the estimation of frequency and error bars with random anisotropic parameters. Here we show the detailed fitting result in fig. 5, where the initial temperature and external magnetic field is $\beta J = 0.04$, $\bar{J} = 0$ and $h/J = 10$.

The choice of the fitting region is naturally uncertain. First, the theoretical prediction of the oscillation frequency acquires the assumption of small $m(t)$, but the small $m(t)$ time region is not uniquely defined. Secondly, especially for ED numerics, finite system size leads to

untrustable results in the late time limit. Therefore it is reasonable only to consider the early time when fitting the ED numerics, which also introduces uncertainty of fitting region.

The error bars of the fitting frequency arise from such uncertainty. To quantify such error, we separately consider two different numerical approaches.

- For the large-$N$ data, we choose the time region begins at different values: $t_{\text{begin}}J = 0, 0.05, 0.1, \ldots, 1.2$, and ends at the same point $t_{\text{end}}J = 10.0$. We fit these frequencies and take the standard deviation as the error bars.

- For the ED data, we choose the time region begins at different values: $t_{\text{begin}}J = 0, 0.08, 0.16, \ldots, 0.4$, and ends at also different points $t_{\text{end}}J = 3.0, 4.0, \cdots 9.0$. We take a combination of these beginning and ending points and take the standard deviation among these frequencies as the error bars.

We summarize the detailed fitting region in table 1. Notice that for the ED approach, we need a larger ending fitting time for the small frequency cases, but in other cases, we choose a smaller ending fitting time to avoid the finite $N$ effect.

Table 1: The detailed parameters and result for each case. $(\xi_1, \xi_2, \xi_3)$ means anisotropic parameters. $\Omega_{\text{KB}}, \Omega_{\text{ED}}, \Omega_{\text{Th}}$ are the fitting frequencies from Kadanoff-Baym numerics, ED numerics, and theoretical prediction. $\sigma_{\Omega_{\text{KB}}}, \sigma_{\Omega_{\text{ED}}}$ are the standard deviations for KB and ED numerics. $[t_{\text{begin,KB}}J, t_{\text{end,KB}}J]$ and $[t_{\text{begin,ED}}J, t_{\text{end,ED}}J]$ are the fit region which lead to the $\Omega_{\text{KB}}, \Omega_{\text{ED}}$ numerics data of the main text fig. 3.

| Cases | 1 | 2 | 3 | 4 | 5 | 6 | 7 | 8 | 9 | 10 | 11 | 12 | 13 | 14 | 15 | 16 | 17 | 18 | 19 |
|---|---|---|---|---|---|---|---|---|---|---|---|---|---|---|---|---|---|---|---|
| $\xi_1$ | -1.46 | 1.47 | 0.36 | 1.79 | -0.8 | -0.51 | 0.25 | -1.58 | -0.01 | 0.09 | 1.51 | 0.25 | -1.49 | 0.89 | -0.73 | 0.56 | -1.83 | -1.15 | 1.31 |
| $\xi_2$ | 1.89 | -1.78 | -1.66 | -1.68 | 1.07 | 0.97 | 0.08 | 1.05 | -0.29 | -1.54 | -0.22 | 0.06 | -1.27 | -0.1 | 1.89 | 0.58 | -0.97 | 1.23 | -1.68 |
| $\xi_3$ | -1.25 | 0.27 | 0.88 | 0.99 | -0.76 | -1.51 | -1.42 | -1.88 | 1.34 | 0.18 | 1.54 | 0.81 | 0.83 | -1.66 | 0.22 | 0.93 | -1.04 | 1.27 | -1.31 |
| $\Omega_{\text{KB}}$ | 3.34 | 1.89 | 2.73 | 2.76 | 1.91 | 2.66 | 1.47 | 3.10 | 1.69 | 1.76 | 1.43 | 0.60 | 2.10 | 1.27 | 1.36 | 0.01 | 0.00 | 0.00 | 0.00 |
| $\sigma_{\Omega_{\text{KB}}}$ | 0.19 | 0.17 | 0.13 | 0.20 | 0.12 | 0.13 | 0.10 | 0.20 | 0.10 | 0.12 | 0.12 | 0.03 | 0.16 | 0.10 | 0.11 | 0.01 | 0.00 | 0.00 | 0.01 |
| $\Omega_{\text{ED}}$ | 3.74 | 1.58 | 3.25 | 2.73 | 2.12 | 3.13 | 1.38 | 3.17 | 1.80 | 1.76 | 0.84 | 0.40 | 2.10 | 0.93 | 0.93 | 0.02 | 0.00 | 0.00 | 0.02 |
| $\sigma_{\Omega_{\text{ED}}}$ | 0.59 | 0.46 | 0.31 | 0.52 | 0.17 | 0.30 | 0.05 | 0.49 | 0.11 | 0.10 | 0.33 | 0.16 | 0.23 | 0.08 | 0.16 | 0.05 | 0.04 | 0.04 | 0.04 |
| $\Omega_{\text{Th}}$ | 3.53 | 1.73 | 3.04 | 2.69 | 2.08 | 2.97 | 1.55 | 3.17 | 1.85 | 1.87 | 1.22 | 0.63 | 2.07 | 1.14 | 1.19 | 0.00 | 0.00 | 0.00 | 0.00 |
| $t_{\text{begin,KB}}J$ | 0.75 | 0.75 | 0.75 | 0.75 | 0.75 | 0.75 | 0.75 | 0.75 | 0.75 | 0.75 | 0.75 | 0.75 | 0.75 | 0.75 | 0.75 | 0.75 | 0.75 | 0.75 | 0.75 |
| $t_{\text{end,KB}}J$ | 10 | 10 | 10 | 10 | 10 | 10 | 10 | 10 | 10 | 10 | 10 | 10 | 10 | 10 | 10 | 10 | 10 | 10 | 10 |
| $t_{\text{begin,ED}}J$ | 0.15 | 0.15 | 0.15 | 0.15 | 0.15 | 0.15 | 0.15 | 0.15 | 0.15 | 0.15 | 0.15 | 0.15 | 0.15 | 0.15 | 0.15 | 0.15 | 0.15 | 0.15 | 0.15 |
| $t_{\text{end,ED}}J$ | 3.5 | 3.5 | 3.5 | 3.5 | 3.5 | 3.5 | 3.5 | 3.5 | 3.5 | 3.5 | 3.5 | 9 | 3.5 | 9 | 9 | 9 | 9 | 9 | 9 |

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
