# Peer review of "Universal Aspects of High-Temperature Relaxation Dynamics in Random Spin Models"

_SciPost Physics, doi:SciPost Phys. 18, 120 (2025)_

## Round 1 · Author Response

Responses to Referee 1's Comments:
We thank the referee for reviewing our manuscript. The referee noted that "the scope is relatively narrow," but we aim to demonstrate that our work has broader implications for the study of dynamics of quantum many-body systems. In particular, we emphasize that our theoretical predictions have recently beendirectly verified by solid-state NMR experiments [Nature Physics 20, 1966–1972 (2024)], published in Nature Physics. We offer the following additional remarks:

Firstly, as indicated in the main text, the randomly interacting spin model, under certain approximations, captures much of the physics observed in experimental systems. These include cold molecules [Nature 501, 521–525 (2013), Phys. Rev. Lett. 113, 195302 (2014)], NV centers [Phys. Rev. Lett. 118, 093601 (2017), Phys. Rev. Lett. 121, 023601 (2018)], trapped fermions [Sci. Adv. 5, eaax1568 (2019)], Rydberg atoms [Nature 491, 87–91 (2012), Phys. Rev. X 11, 011011 (2021)], high-spin atoms [Phys. Rev. Lett. 125, 143401 (2020)], and solid-state NMR systems [Phys. Rev. Lett.97, 150503 (2006), Phys. Rev. A 75, 042305 (2007)].

Secondly, while the two-point correlator function is one of the simplest observables, it provides significant insights into fascinating many-body physics and serves as a valuable testbed. To date, investigations of quantum dynamics have been central to numerous experiments, including Ramsey contrast detection in NV center systems [Phys. Rev. Lett. 130, 210403 (2023)] and polar molecular systems [Nature 633, 332–337], as well as free induction decay in solid-state NMR systems [Nature Physics 20, 1966–1972 (2024)]. These studies have successfully predicted the correct scaling behavior in decay rates.

Thirdly, our framework provides an analytical expression for quantum spin dynamics that cannot be achieved through purely numerical methods such as exact diagonalization or semi-classical mean-field approximations. For a long time, it has been observed in NMR experiments that the two-point spin correlator exhibits either oscillatory decay or monotonic decay. However, the underlying physical mechanism and a complete determination of the phase diagram have remained unexplored [Phys. Rev. Lett. 101, 067601 (2008)]. Our framework addresses these gaps and can be further extended to incorporate more detailed features of experimental systems.

Responses to Prof. Subir Sachdev's Comments:
We sincerely thank Prof. Subir Sachdev for his efforts in reviewing our manuscript and providing valuable suggestions. We are greatly encouraged by his report, which recognizes our manuscript as ``well written'' and ``containing important new results on the non-equilibrium dynamics of a strongly interacting quantum system at high temperature''.

We have revised the abstract as follows: "This result is derived from the Kadanoff-Baym equation under the melon diagram approximation, which is consistent with numerical solutions. Furthermore, we verify our theory and approximation using exact diagonalization, albeit for a small system size of $N=8$." Similar revisions have been made to the introduction and discussion sections as well.

---

## Round 1 · List of Changes

In response to Comment 2-1. We revise discussions in the abstract and introduction as: "This result is derived from the Kadanoff-Baym equation under the melon diagram approximation, which is consistent with numerical solutions. Furthermore, we verify our theory and approximation using exact diagonalization, albeit for a small system size of $N=8$."

---

## Editorial Decision

published